# Nullspace of Vision Transformers and What Does it Tell Us?

## Abstract

Nullspace of a linear mapping is the subspace which is mapped to the zero vector. For a linear map, adding an element of the nullspace to its input has no effect on the output of the mapping. We position this work as an exposition towards answering one simple question, "Does a vision transformer have a non-trivial nullspace?" If TRUE, this would imply that adding elements from this non-trivial nullspace to an input will have no effect on the output of the network. This finding can eventually lead us closer to understanding the generalization properties of vision transformers. In this paper, we first demonstrate that provably a non-trivial nullspace exists for a particular class of vision transformers. This proof is drawn by simply computing the nullspace of the patch embedding matrices. We extend this idea to the non-linear layers of the vision transformer and show that it is possible to learn a non-linear counterpart to the nullspace via simple optimisations for any vision transformer. Subsequently, we perform studies to understand robustness properties of ViTs under nullspace noise. Under robustness, we investigate prediction stability, and fooling properties (network and interpretation) of the noise. Lastly, we provide image watermarking as an application of nullspace noise.

## 1 Introduction

In recent years, deep learning models have seen tremendous success. So much so, that now these models are the standards for tasks ranging from image recognition (He et al., 2016; Dosovitskiy et al., 2021; Wortsman et al., 2022), object detection (Dai et al., 2021a; Guan et al., 2022), scene segmentation (Fang et al., 2021; Zhang et al., 2022b), language translation (Devlin et al., 2019; Xue et al., 2022), speech recognition (Nagrani et al., 2021; Chen et al.) among many others. We are now observing their application to various novel problems such as predicting protein structures (Jumper et al., 2021), generative modelling (Dhariwal & Nichol, 2021), autonomous driving (Grigorescu et al., 2020), solving differential equations (Lample & Charton, 2020) to name a few. It is only fair to assume that we shall be witnessing an accelerated adoption of deep learning models into our daily lives as the years go by.

In computer vision, most of the architecture types can be broadly classified into two groups: convolution neural networks (CNNs) and transformers based on their building blocks. CNNs have gained overwhelming popularity since their state of the art redefining performance (Ciregan et al., 2012; Krizhevsky et al., 2012) on the ImageNet challenge (Russakovsky et al., 2015). The main characteristic of CNNs is the use of trainable kernels to perform convolution operation in a strided fashion on the inputs (LeCun et al., 1989; O'Shea & Nash, 2015). On the contrary, transformer based architectures like Vision Transformers (ViTs) (Dosovitskiy et al., 2021) are more recent inventions compared to the CNNs. Key aspect of transformer based architectures is the adaptation and utilisation of self/cross-attention modules (Vaswani et al., 2017).

In the short span of three years, transformer based classes of vision models have gained tremendous popularity. They compete with CNNs on various computer vision tasks (Touvron et al., 2021; Carion et al., 2020; Arnab et al., 2021). The focus of our work is a specific architecture, the Vision Transformer. It processes an input image by first splitting it into several non-overlapping patches followed by a linear projection which is then processed by a vanilla transformer. Since their introduction in 2020, ViTs have been the source of inspiration for several recent novel architectures (Ali et al., 2021; Li et al., 2022b; Liu et al., 2021). Also, researchers have made multitude of recent

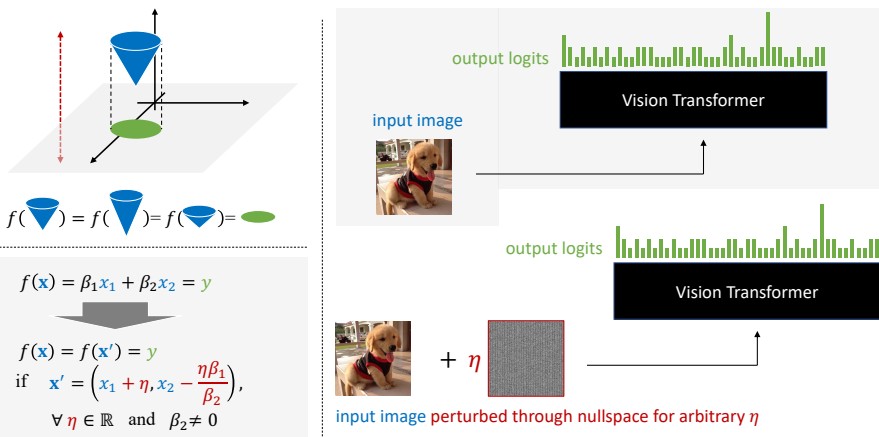

Figure 1: **An illustration of the nullspace in three cases (projection function case, left top; linear function case, left bottom; vision transformer case, right)**. For the functions in these three cases, there exists some nullspace, and the output of the function with respect to the input will remain the same no matter how much perturbation is introduced to the input along the nullspace. Also, the nullspace is function-specific (model-specific) and will not vary for different samples.

discoveries about the properties of transformer based architectures with ViTs as the focal point of their studies (Naseer et al., 2021; Mahmood et al., 2021; Minderer et al., 2021). This decision to prioritize ViTs stems from the simplicity of ViTs resulting in fewer engineering hurdles and high flexibility and versatility which leads to wider adoption across domains.

Despite the architecture is still relevatively new, the community has made several insightful findings along the investigation of the working mechanisms of it. For example, Naseer et al. (2021) showed that ViTs are robust to occlusions, input perturbations and domain shifts, as well as bearing a lower reliance on local textures as compared to that of CNNs as showed earlier by Geirhos et al. (2018). These findings were also corroborated in a recent work (Zhang et al., 2022a). Further, Zhou et al. (2022) verified that the robustness of ViTs to several corruptions is primarily due to the self-attention blocks. However, a recent finding by Pinto et al. (2021) attributes relatively better performance of ViTs on out-of-distribution generalisation to flawed comparison of models solely based on number of parameters. On a related topic, along the study of adversarial robustness, Mahmood et al. (2021) found that, unlike CNNs, adversarial examples for ViTs had lower transferability across architectures, which inspired them to build a more robust ensemble. On the other hand, as per Bai et al. (2021), the improved generalisation of ViTs is benefited by the self-attention framework employed by the network. They also reported ViTs are as vulnerable to adversarial attacks (Goodfellow et al., 2015; Moosavi-Dezfooli et al., 2016) as their CNN counterparts under a fair evaluation. Moreover, they found no evidence suggesting that larger datasets helped ViTs more than CNNs in improving generalisation. These two findings are in conflict with earlier works inspecting ViTs (Bhojanapalli et al., 2021; Shao et al., 2021). On the aspect of neural network calibration (Guo et al., 2017), Minderer et al. (2021) observed that recent architectures, the likes of ViTs, were much better calibrated in their prediction scores. Finally, the architectural differences between CNNs and ViTs naturally result in the differences of the inductive biases (Raghu et al., 2021), and, in comparison to CNNs, ViTs lack inductive biases for local structure (edges, corners). To remedy this, Dai et al. (2021b); Xu et al. (2021); Li et al. (2022a) proposed solutions resort to either large amount of training data or inclusion of convolution layers.

Though no single architecture type comes off as the clear winner, it is fair to believe that ViTs are in general the primary focus of the community when studying transformer architectures in vision. Sharing the motivation to explore and further our understanding of ViTs akin to previous works, in this paper, we aim to highlight their untouched aspect: the `nullspace` of vision transformers. Nullspace of a linear map $f : \mathcal{X} \rightarrow \mathcal{Y}$ is the subspace of $\mathcal{X}$ which is mapped to $\mathbf{0}$. Formally, $\Delta = \{\mathbf{x}|f(\mathbf{x}) = \mathbf{0}\}$.

**Why is nullspace important?** A seemingly innocent concept of nullspace describes a very interesting property of ViTs. If such a subspace exists for ViTs, then it would imply that the network is inherently robust to certain types of input perturbations (or corruptions). That is, simply sampling

a perturbation from this nullspace and adding to the input will have no effect on the output of the network. Moreover, in theory, we can synthesise infinite such perturbations by simply varying the weights for the basis vectors of this subspace. Figure 1 provides a pictorial representation of the concept of nullspace for different mathematical formulations.

We begin the exploratory study of nullspace of ViTs by first providing a formal introduction to nullspace in section 2. In section 3, we show that nullspace exists for certain vision transformers. We introduce approximate nullspace noise for non-linear components of ViTs in section 4. We revisit the idea of robustness in section 5 and evaluate ViTs and CNNs on different nullspace noises. In section 6, as an interesting application of nullspace, we explore the utility of nullspace perturbations for image watermarking. Extending the formulation of image watermarking, we show that it is possible to generate samples which can fool the model and several interpretability methods (section 7). In section 8, we discuss related works in the field of neural networks and nullspace. Lastly, we discuss the implications of our findings (section 9) and conclude our work (section 10).

## 2 PRIMER: NULLSPACE

Given a input $\mathbf{x} \in \mathbb{R}^{1 \times p}$ and a model vector $\beta \in \mathbb{R}^{p \times k}$, we can build a simple regression model

$$\mathbf{y} = \mathbf{x}\beta \tag{1}$$

where we call $y$ the responses and $\mathbf{y} \in \mathbb{R}^{1 \times k}$. Interestingly, given a fixed $\beta$, we can find vectors $\Delta$, such that

$$\mathbf{u}\beta = \mathbf{0}, \forall \mathbf{u} \in \Delta \tag{2}$$

In that case, for a given model with $\beta$, the output of $\mathbf{x}$ and $\mathbf{x} + \Delta$ will be exactly the same. $\Delta$ is called the nullspace of $\beta$. For a linear mapping such as $\beta$, where the domain is a vector space, nullspace is a subspace and satisfies all the required axioms:

- Zero element of $\mathbb{R}^{1 \times p} \in \Delta$. This is the true as $\mathbf{0}\beta = \mathbf{0}$.

- $\Delta$ is closed under vector addition. $\mathbf{u}+\mathbf{v} \in \Delta \ \forall \mathbf{u},\mathbf{v} \in \Delta$. This is true as $(\mathbf{u}+\mathbf{v})\beta = \mathbf{u}\beta^{\mathbf{0}}+\mathbf{v}\beta^{\mathbf{0}} = \mathbf{0}$.

- Closed under multiplication with scalar. $c\mathbf{u} \in \Delta \ \forall \ c \in \mathbb{R}$ and $\mathbf{u} \in \Delta$. This is true as $(c\mathbf{u})\beta = c(\mathbf{u}\beta) = \mathbf{0}$.

A trivial nullspace, $\Delta = \{\mathbf{0}\}$, always exists for a linear mapping as described in equation 1. An alternate way is to interpret nullspace is to view it as a set of solutions to system of linear equations as described by equation 1. This imples that $\mathbf{0}$ is always a solution to the said equation. As the number of solutions to a system of linear equations can vary, the nullspace for a mapping can be trivial, non-trivial, or does not exist.

**How to solve for $\Delta$?** Given a matrix, finding its nullspace has now become an ubiquitous exercise in mathematical textbooks. We refer the readers to excellent resources on this topic (Kwak & Hong, 2004; Strang, 2009b;a). In our work, we rely on the Python package of *Numpy* for finding the nullspace of any given matrix (Harris et al., 2020).

## 3 EXACT NULLSPACE

In this section, we first review the working of a vision transformer. Following it, we demonstrate the existence of nullspace for ViTs.

### 3.1 VISION TRANSFORMER

Vision transformer as introduced by Dosovitskiy et al. (2021) is a function $f_\omega$ with $\omega$ as the trainable weights. The function takes as input an image $\mathbf{x} \in \mathcal{X}^{c \times h \times w}$ and outputs a classification response $\mathbf{y} \in \mathcal{Y}^k$ over $k$ categories. $c$ is the number of channels (typically 3 for red, green, and blue), $h$, $w$ correspond to height and width of the input image. This neural network function can be broken down into composition of 3 stages, namely:

- *patch embedding stage*, $f_\theta : \mathcal{X}^{c \times r \times r} \to \mathcal{U}^d$. This steps projects the input image patch of pre-determined dimensions $c$, $r$ and $r$ to a one-dimensional embedding of length $d$. It is ensured that patches have no overlaps between them. Hence, the number of such non-overlapping patches generated from the input image are $m = \frac{h \times w}{r^2}$.

- *self-attention stage*, $f_\phi : \mathcal{U}^{(m+1) \times d} \to \mathcal{V}^{(m+1) \times d}$. In the next step, the generated patch embeddings are passed through layers of self-attention modules to process long range interactions amongst them. Apart from the $m$ patch embeddings an additional embedding in form of `cls` token is utilised in this step.

- *classification stage*, $f_\psi : \mathcal{V}^d \to \mathcal{Y}^k$. The final step is to perform the $k$-way classification. For this, we simply keep the processed encoding corresponding to `cls` token and project it through a linear classification layer.

This breakdown can also be illustrated as:

$$\mathbf{x}_i^{c \times r \times r} \underset{f_\theta}{\longrightarrow} \mathbf{u}_i \qquad [\mathbf{u}_{cls}\mathbf{u}_0 \ldots \mathbf{u}_m] \underset{f_\phi}{\longrightarrow} [\mathbf{v}_{cls}\mathbf{v}_0 \ldots \mathbf{v}_m] \qquad \mathbf{v}_{cls} \underset{f_\psi}{\longrightarrow} [y_0 y_1 \cdots y_k] \quad (3)$$

## 3.2 Nullspace in Transformer

Now we consider the first layer of the transformers i.e. the *patch embedding* layer, $f_\theta$. We consider $\theta$ with $d$ kernels, and each has $r \times r$ weights. Then for a input patch $\mathbf{x} \in \mathbb{R}^{1 \times r \times r}$, we have the representation (embeddings) learned from that patch as

$$\mathbf{y} = \mathbf{x}\theta.$$

As per the discussion in section 2, we can find $\Delta_\theta$, the nullspace for $f_\theta$. Let $B_\theta = \{\mathbf{b}_1, \mathbf{b}_2, \ldots \mathbf{b}_d\}$ be the $d$ basis vectors for this nullspace. As per the axioms described in section 2, we can sample an element from $\Delta_\theta$ as:

$$\mathbf{v}_\theta = \lambda_1 \mathbf{b}_1 + \lambda_2 \mathbf{b}_2 + \cdots + \lambda_d \mathbf{b}_d. \tag{4}$$

This would imply that the output of the patch embedding will effectively remain preserved, $f(\mathbf{x} + \mathbf{v}_\theta) = f(\mathbf{x})$. Since the output after the first layer remains unaffected the final output of the classification remains unchanged. In this manner, nullspace of patch embedding also serves as a subset of the nullspace of the vision transformer.

We refer to $\mathbf{v}_\theta$ as **nullspace noise** or **nullspace perturbation**. It is important to note that nullspace noise only depends on the patch embeddings weights and is independent of the input. As a result, the same noise can be added to any input without impacting the final output.

**Does a nullspace always exist for ViTs?** Specific values of the weights determine whether a non-trivial nullspace exists or not, hence, we require a case by case analysis of ViT architectures. However, as a general rule of thumb, ViT configurations with $d < r^2$ will always have a non-trivial nullspace (as per the rank-nullity theorem[1]). Experimentally, we verified that for ImageNet (Russakovsky et al., 2015) pretrained ViTs, standard configurations of ViT-Large, ViT-Base, ViT-Small with patch sizes of $32 \times 32$ have a non-trivial nullspace. As the embedding dimensions for these networks are ordered as $d_{\text{small}} < d_{\text{base}} < d_{\text{large}}$, per the rank-nullity theorem for the patch size 1024 we get the dimensions of the nullspace in the order $\Delta_{\theta,\text{small}} > \Delta_{\theta,\text{base}} > \Delta_{\theta,\text{large}}$. The ordering is corroborated by the experimental results.

## 4 Approximate Nullspace Noise

In the previous section we have demonstrated that a non-trivial nullspace exists for the patch embedding and hence the entire vision transformer, in this section we move further down the structure of ViT and investigate if a nullspace exists for the self-attention stage, and thus whether we can enlarge the nullspace set of the entire vision transformer. To recall, self-attention stage applies a series of *QKV* attention operations followed by normalisation and non-linear transformations. The overall operation is thus *non-linear* which implies the concept of nullspace for $f_\phi$ is ill-defined. This is because even if such a set exists where $f_\phi(\mathbf{v}) = \mathbf{0}$, $\forall \mathbf{v} \in \Delta_\phi$, this set is no longer guaranteed to

---

[1]Rank-nullity theorem asserts that the sum of rank of a matrix and dimensions of its nullspace should be equal to the dimensions of the said matrix.

be closed under vector addition and scalar multiplication. Hence, we search for an alternative to nullspace in the non-linear case.

Regardless, we can still attempt to preserve the axiom of most interest to us, *closeness under vector addition*. We attempt this through formulating the concept into adding arbitrary noise without disturbing the output of a function. Therefore, instead of looking for a vector space, we can instead search for a set with the following property:

$$\Delta_\phi = \{\mathbf{v}|f(\mathbf{u} + \mathbf{v}) = f(\mathbf{u}) \,\forall \mathbf{u} \in \mathcal{U}\}. \tag{5}$$

i.e. adding elements from $\Delta_\phi$ to the input of $f_\phi$ has no impact on the output of the attention modules.

However, analytically finding $\Delta_\phi$ is still challenging. Thus, we seek for a feasible alternative: We further relax the searching criteria from a set to an individual element of this set, $\mathbf{v}_\phi$. $\mathbf{v}_\phi$, is a perturbation, when introduced to any input of $f_\phi$, does not impact its output. Therefore, we can turn the searching for $\mathbf{v}_\phi$ into the following optimisation process that searches for $\tilde{\mathbf{v}}_\phi{}^2$ as an approximation of $\mathbf{v}_\phi$:

$$\mathcal{L}_\phi = \|f_\psi(f_\phi(\mathbf{u} + \tilde{\mathbf{v}}_\phi)[0]) - f_\psi(f_\phi(\mathbf{u})[0])\|_p \tag{6}$$

where, indexing 0 highlights the use of `cls` token in the loss function, and the normalisation can be any $\ell_p$ norm. The above loss function tries to minimize the $\ell_p$ norm between the predicted logits with and without $\mathbf{v}_\phi$. Alongside the self-attention stage we have also incorporated the classification stage into the loss, since our the target of our study is to minimize the impact on the *final* output of the network. To minimise the loss, we use a simple gradient descent based procedure to minimise the loss. A PyTorch styled pseudo-code along with the implementation details are in the appendix (section A.1). We initialise the $\tilde{\mathbf{v}}_\phi$ as a noise sampled from $Uniform(-lim, lim)$ for the minimisation of loss (also referred to as sampling limit).

To quantitatively evaluate learnt $\tilde{\mathbf{v}}_\phi$, in figure 2 we report the percentage of matching classifications with and without learnt nullspace noise along with mean squared error computed at the output logits. We consider a prediction to be matched if the assigned category for an input is the same with and without adding the perturbation. The results indicate that increasing $lim$ decreases the percentage of matching predictions and increases the mse. This is expected as the optimisation converges towards a local minima with relatively larger noise values. We also show that the learnt values of $\tilde{\mathbf{v}}_\phi$ is important by also reporting the results after shuffling the noise values prior to adding it to the input (depicted with $---$). Shuffling the values indicate that the learnt values of $\tilde{\mathbf{v}}_\phi$ are meaningful and the noise does not just comprise of negligible values. As the dataset, we have used ImageNette (Howard, 2018) which is a subset of ImageNet dataset consisting of 10 categories.

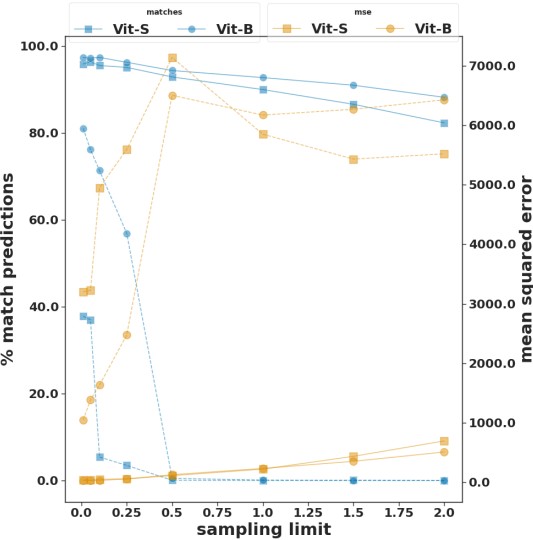

Figure 2: $\tilde{\mathbf{v}}_\phi$ **evaluation**.

We learnt $\tilde{\mathbf{v}}_\phi$ on the training dataset ($\approx 9500$ images) and performed evaluation on the validation set ($\approx 4000$ images).

# 5 STUDY: UNDERSTANDING ROBUSTNESS WITH NULLSPACE NOISE

## 5.1 BACKGROUND

Deep neural networks tend to be fragile. It is widely known that simple perturbations to the input can result in unintended outputs. If these perturbations are generated adversarially, the output of a network misclassifies the sample even if visibly nothing much has changed in the input image

---

[2]Though $\tilde{\mathbf{v}}_\phi$ is an approximation of the nullspace noise, we refer to it as nullspace noise in our work and clearly highlight the demarcation where required.

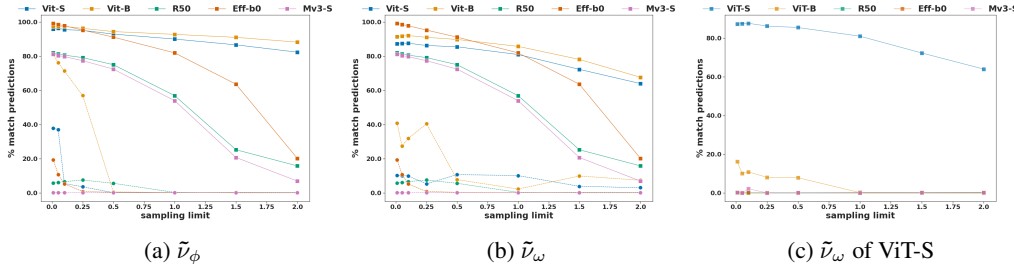

(a) $\tilde{\nu}_\phi$        (b) $\tilde{\nu}_\omega$        (c) $\tilde{\nu}_\omega$ of ViT-S

Figure 3: **Performance under different noises.** Figures (a), (b) show the performance degradation under approximated noise learnt at encoder and input levels. For both, the noise computation is same for CNNs. In figure (c), we show the performance under exact nullspace noise from ViT-S applied to other networks.

(Goodfellow et al., 2015; Nguyen et al., 2015; Moosavi-Dezfooli et al., 2016; Ilyas et al., 2019). Majority of adversarial attacks are generated per input sample, i.e. given an input image the adversarial noise is synthesised curated for the input sample. There are also approaches which have attempted to overcome this limitation where the adversarial noise is learnt per model (Moosavi-Dezfooli et al., 2017; Liu et al., 2017; Brown et al., 2017). A key difference between adversarial perturbations and nullspace noise is that with the addition of the input-independent nullspace noise, we aim to maintain the final prediction of the model.

Departing from learning specific input perturbations, adding noise in form of 'corruptions' has also been a theme for testing robustness under data distribution shifts (Hendrycks & Dietterich, 2018; Geirhos et al., 2018; Recht et al., 2019; Wang et al., 2019; Hendrycks et al., 2021a;b). These evaluations tend to highlight lack of generalisation prowess in deep neural networks as the testing data diverges slightly (or at times heavily) from the distribution of training data. Many works have benchmarked and attempted to mitigate the deterioration in performance under such shifts (Taori et al., 2020). As opposed to nullspace noise, these corruptions are crafted manually to bring forth a potential pitfall of the model. Hence, a particular corruption can arbitrarily fail or succeed on any given image.

In this section, we aim to understand robustness of vision transformers focused at its nullspaces. We pose robustness w.r.t nullspace as the question **"How robust are different architectures to nullspace noises."**

## 5.2 EXPERIMENT: NOISE ROBUSTNESS

For vision transformers, we have computed two types of noises, $\mathbf{v}_\theta$ and $\tilde{\mathbf{v}}_\phi$. To further demonstrate the robustness advantages of ViT, we also need to calculate the nullspace of CNNs for comparison. However, opposed to ViTs, CNNs uses overlapping strided convolutions instead of patch embedding. Application of multiple kernels on overlapping input patches makes it intractable to determine if a nullspace for a CNN exists at the very first layer, since any nullspace, if exists, must be shared by multiple kernels. The candidate pool is further constrained by strided convolutions. Therefore, instead of analytically solving for the nullspace, we have to approximate it as we did for the attention stage of ViT. For comparison, we mainly perform the following experiments:

- Compare performance of different architectures under approximate nullspace noise for encoders. This will help us contrast the *goodness* of nullspace noise of ViTs to CNNs.

- Study the impact of transfer of noise from a ViT to other ViT and CNN models. This will allow us to understand the extent to which different architectures behave under noise transfer.

## 5.3 RESULTS

In figure 3(a), we compare the performance of ViTs under noise learnt at $f_\phi$ level. In comparison, for CNNs the noise is learnt for the entire network. Under similar sampling limit we measure the performance variations due to non-linear encoders for different architectures. We can observe that compared to R50 (ResNet-50), Eff-b0 (Efficientnet-b0) (Tan & Le, 2019) and Mv3-S(MobilenetV3-Small) (Howard et al., 2019), ViTs perform better under $\tilde{\mathbf{v}}$ throughout different sampling limit. For

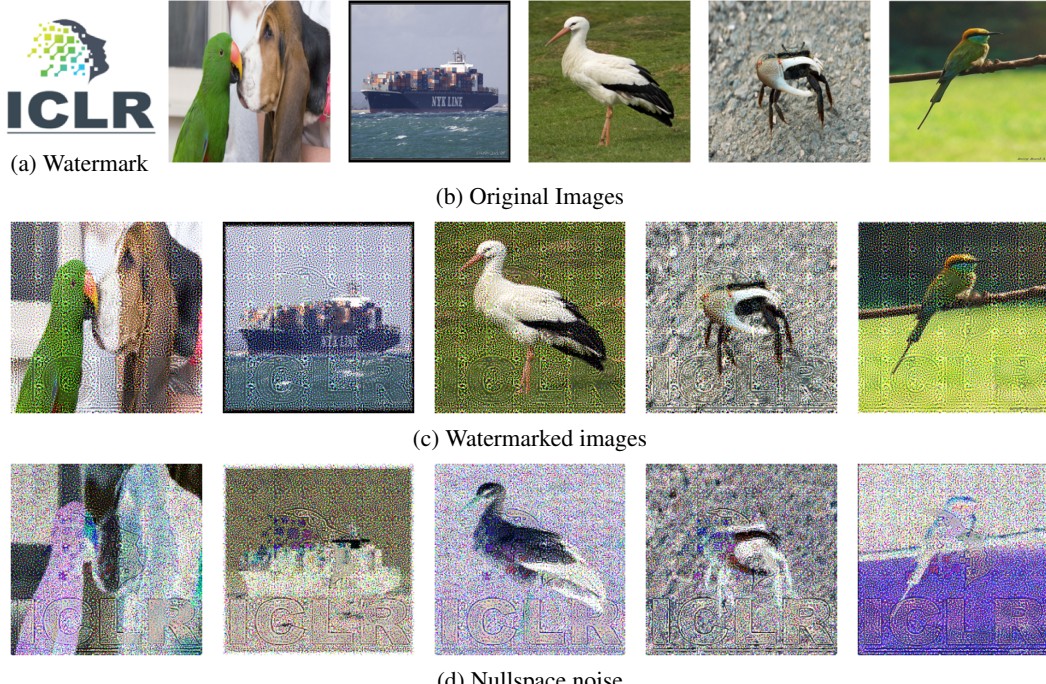

Figure 4: **Watermark superposition using nullspace basis vectors.** Figures (a) and (b) correspond to the watermark image and the original images. The watermarked images in (c) are generated using the nullspace basis vectors, thus images in (c) will trigger the model for the same set of outputs as in (b). The additive nullspace noises (d) show the overlaid watermark content. Best viewed digitally.

CNNs, Eff-b0 performs better at low limit (0.01, 0.05) but the performance drastically depletes at higher sampling limit. Poor performance of CNNs compared to ViTs' non-linear counterpart supports our hypothesis that nullspace noise for CNNs is hard to construct.

In figure 3(b), we learn $\tilde{\mathbf{v}}_\omega$ for the entire network from the input level ($\omega$ represents the model parameters). In this manner, we perform a fair evaluation at the network level. We see a similar trend to what we observed for figure 3(a). For ViTs, we observe that for noise learnt at the first layer results in relatively worse performance at larger sampling limits than for $\tilde{\mathbf{v}}_\phi$. Overall, from figures 3 (a,b) we can conclude that ViTs are more robust to approximate nullspace noises compared to their CNN counterparts.

In figure 3(c) we report results of ViT-S' $\tilde{\mathbf{v}}_\omega$ being applied to the inputs for ViT-Base and CNNs. ViT-S naturally attains the best performance at all levels. ViT-B responds better than CNNs to the transferred noise and performs better than random choice. It can be attributed to architectural similarities between the two ViTs. For the CNNs, we record significantly lower performance. Lack of effective nullspace noise can play a key role here. Overall, we observe very low transferability of noise between architectures.

## 6 APPLICATION: OUTPUT-PRESERVED IMAGE WATERMARKING

Watermarking as image, usually used to convey ownership information or verify content of the data, has been studied extensively (Wolfgang & Delp, 1996; Potdar et al., 2005; Al-Haj, 2007; Bhat et al., 2010). A watermark can be either imperceptible or perceptible. and perceptible watermarking applies a noticeable marker to convey the protected nature of the data (Berghel, 1998). In this section, we explore to utilize nullspace noise to apply a perceptible watermark on the image which does not alter the test-time forward process.

Figure 4 shows an example watermarking approach using the nullspace noise. Here, we watermark a source image with a watermark. The resulting modified image, figure (c), attains the final predictions close to the original image. We observe instances of the text appearing in the modified image.

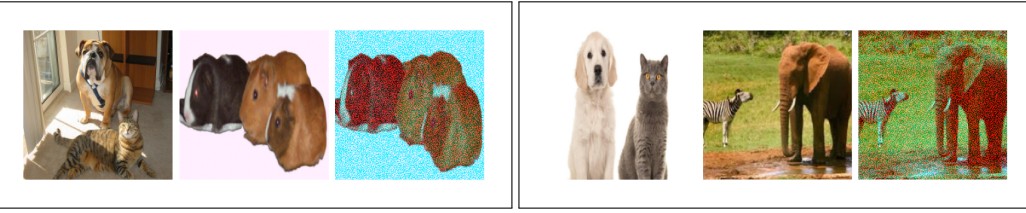

(a) Triplet of Source, target and transformed images

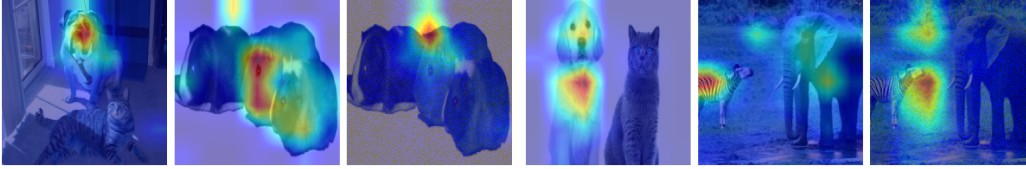

(b) Saliency maps for the corresponding images from the row above.

Figure 5: **Targeted nullspace noise.** Transformed images appear visually as target images but are interpreted as source images by the model. The equivalence between source and transformed images is not only in terms of the final predictions but also in the interpretability maps depicted in (b).

**Method details:** With basis vectors of the nullspace, we can construct a watermark to be overlaid on the original image without affecting the output of the network. Given a source (user's image) and a target image (watermark), we simply need to estimate the scalar parameters corresponding to the basis vectors to satisfy $\sum_{i=0}^{i<m} \mathbf{e}_i \lambda_i = \mathbf{v}_\theta \approx \Delta\mathbf{x}_j$. $\mathbf{e}_i$ are the basis vectors for the nullspace, $\lambda_i$ are their corresponding scalar co-efficients which are to be determined and $\Delta\mathbf{x}_j$ is the changed required to convert $j^{\text{th}}$ original image patch to $j^{\text{th}}$ watermark image patch. This can be achieved through a constrained optimisation of the following form:

$$\min\|\Delta\mathbf{x}_j - \sum_{i=0}^{i<m} \mathbf{e}_i \lambda_i\|_p, \quad \text{subject to} \quad 0 \geq \mathbf{x}_j + \sum_i \mathbf{e}_i \lambda_i \leq 1. \tag{7}$$

The constraints restrict the pixel values of the altered image to be within $[0, 1]$. The optimisation can be solved using out of the box Sequential Least Squares Programming optimizer (Nocedal & Wright, 2006). In the appendix (A.2), we report the quantitative results of the network on watermarked images.

## 7 STUDY: UNCONSTRAINED TARGETED NOISE

Using watermarking allows us to impose information about the watermark image on any source image. However, the output will still be slightly altered potentially due to (i) limited expressiveness imposed due to the nullspace (ii) the constraint within 7. In this section, we remove the constraint and perform a study to understand the impact on a model. By this relaxation, we will show that it is possible to *fool* both the model and interpretation methods.

Similar to the study of adversarial robustness, we consider a model to be fooled if the predicted category differs for an input before and after application of noise. In figure 5(a) we display the qualitative results of synthesising a transformed image using a targeted $\tilde{\mathbf{v}}_\theta$ operation. The transformation is not perfect, however, we can spot that the transformed images are visually similar to target images than source images. Even with this difference in the input space, transformed images and source images are classified into the same category with roughly the same confidence. We deflect the exact details of the optimisation process to the appendix (see section A.3).

As recent studies have shown, fooling can also be extended to the interpretability methods (XAI) Dombrowski et al. (2019) partially due the limitations exposed by recent studies (Dombrowski et al., 2019; Ghorbani et al., 2019; Heo et al., 2019). However, in contrast to these works aiming to fool specific XAI method, our nullspace noise only depends on the model, not the XAI method.

In figure 5(b), we show the interpretability maps as generated by LRP (Chefer et al., 2021). From the figure, we can observe that the heatmaps generated by source and transformed images are iden-

tical whereas, the transformed image heatmaps substantially differ from target images'. Though only reported for LRP, we observed that a similar observation holds across different interpretability approaches. Here, we only presented the results on LRP, as in the context of ViTs, we found the heatmaps from other methods to be lacking (also pointed out by authors of LRP). Visualisations on other interpretability methods are provided in the appendix (see section A.4).

## 8 RELATED WORK

In this section, we keep the discussion limited to related works combining neural networks and nullspaces. To the best of our knowledge, the earliest work investigating neural networks alongside nullspaces corresponds to the study by Goggin et al. (1992). They studied the universal approximation capabilities of a multi-layered perceptrons by comparing the outputs and nullspace of inputs. Through a classical example of *learning XOR* they showed that with the use of hidden layers, an MLP is able to construct a transformation which maps input to targets even if the target happens to be in the nullspace of the input. In a much recent work, Sonoda et al. (2021) mathematically specified the behavior of nullspace, but only for fully connected networks. On the application side, for a continual learning setting, Wang et al. (2021) proposed to map new tasks to the nullspace of the existing tasks. NuSA (Cook et al., 2020) is a framework for outlier detection using the nullspace analysis of a linear layer in a neural network. The idea is to monitor projection magnitude on the nullspace during test and compare it with a threshold. Lastly as a novel architecture, Abdelpakey & Shehata (2020) proposed NullSpaceNet which maps inputs from the same category to a joint-nullspace instead of a feature space.

## 9 DISCUSSION

Before we conclude, we would like to devote a short section for the discussion of the limitation of our findings and other potential technical implications.

**Societal Impact** Although the name nullspace might imply a negative property, we notice the most practical implication of nullspace is to offer explanations to ViT's additional resilience toward minor noises in comparison to CNNs (Section 5). Our results in Section 7 might also be interpreted as a threat, but the perturbations are barely powerful enough to fool the models within the constraint of typical pixel values. Thus, we do not expect our work to introduce any negative societal impacts.

**Applications in Model Patenting** In addition to the applications we discussed, we consider another potential usage of our findings is to patent a ViT after a model is trained, as the nullspace will be unique property of any set of weights of certain ViT architectures. Different from the existing line of research in model watermarking (Adi et al., 2018; Darvish Rouhani et al., 2019; Le Merrer et al., 2020), the patenting through nullspace will not require any additional steps during training, although will face limited usage scenarios in comparison.

**Potential Limitation about the Nullspace Approximation** Different from the nullspace defined in linear algebra, the nullspace of the entire ViT can only be approximated because of the non-linearity in the network architecture. However, it is worthy mentioning that we can still calculate the exact nullspace of ViT if we only consider the patch embedding layer, through which, our results will qualitatively deliver the same message, but with quantitative differences.

## 10 CONCLUSION

In this work, we have explored the concept of nullspaces for vision transformers. By only utilising the linear projection component, we showed that non-trivial nullspaces exist for various configurations of ViTs. Subsequently, we extended this idea to the non-linear component of a vision transformer and demonstrated a simple method of generating approximate nullspace noise. By comparing ViTs with contemporary CNNs, we showed that ViTs are fairly robust to high amount of noise generated via nullspace. From an application perspective we demonstrated that nullspace noises can be readily utilised to impose perceptible watermarks on images without altering the final predictions casted by a network. Lastly, we demonstrate that under an unconstrained setting, nullspace noises can be easily wielded to fool both the network and interpretability methods.

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

# A APPENDIX

## A.1 NULL COMPONENTS

---

**Algorithm 1** Learning $\tilde{\mathbf{v}}_\phi$

---

```
1  # fθ, fφ, fψ
2  # epochs: Training epochs # 500
3  # ε: Learning rate # 0.01
4  # steps: Epochs to reduce the learning rate # [150, 300, 400]
5  # lim: Range to randomly sample initial values for ṽφ # [.01, .05, .1, .25, .5, 1, 1.5, 2]
6
7  ṽφ ~ U(−lim, lim)
8  for epoch in range(epochs):
9      for x in loader: # load a minibatch x of images
10         with torch.no_grad():
11             gt = fψ(fφ(fθ(x)))  # original output logits
12
13         u = fθ(x) + [0; ṽφ]  # introducing noise
14         out = fψ(fφ(u))
15         loss = (out − gt)².sum(dim=−1)^0.5.mean()  # L2 computation
16
17         loss.backward() # back−propagate
18         grad = ṽφ.grad()
19         ṽφ = ṽφ − ε * grad  # gradient step
20
21     if epoch in steps:
22         ε = ε*0.1
23
24  return ṽφ
```

---

## A.2 WATERMARKING IMAGES

As mentioned in the main paper, we use SQSLP for minimising equation 7. We use the implementation provided by SciPY (Virtanen et al., 2020) and run it 5000 iterations. Experimentally, we observed that using $p = 1$ norm provides better watermarking behaviour than $p = 2$. All reported and displayed nullspace noise content is with $p = 1$ unless stated otherwise.

To quantitative assess the robustness of model to the watermarking process, we will have to watermark every image in the dataset. This process requires considerable time and compute to execute. Instead, we perform the evaluation for randomly selected 20 images and compute the % match predictions and absolute difference in the predicted probabilities. We found that $85\%$ of the watermarked images were classified as the source image category. For the mean absolute difference, we compute it between the predicted probabilities for source image category both for the source image and the watermarked image. We observed that the difference in confidence varied $11.63\%$ on an average.

In figure 6 we show the original, $|\mathbf{v}_\theta|$ and the resulting watermark images. The watermark image is the same as one reported in the main paper. Using nullspace watermarking, we notice that shape details are more likely to be transferred than other information from the watermark image.

## A.3 TARGETED NULLSPACE NOISE

Instead of directly minimising the unconstrained equation 7 with huge number of variables ($r \times r \times c$), we manually fix the values of 2 channels (green and blue) and only perform the minimisation to learn the red channel values for the transformed image. This reduces the number of parameter to one-third and also retains a lot of target image information without any loss. This is also the reason why we observe different colored tint for the transformed images. With respect to the implementation, the details are identical to that for watermarking.

## A.4 FOOLING XAI METHODS

In figure 7 we show the saliency maps generated by various XAI methods. Even though the maps generated by methods other than LRP are poor (hard to interpret), we see that the source and transformed respond similarly to these methods.

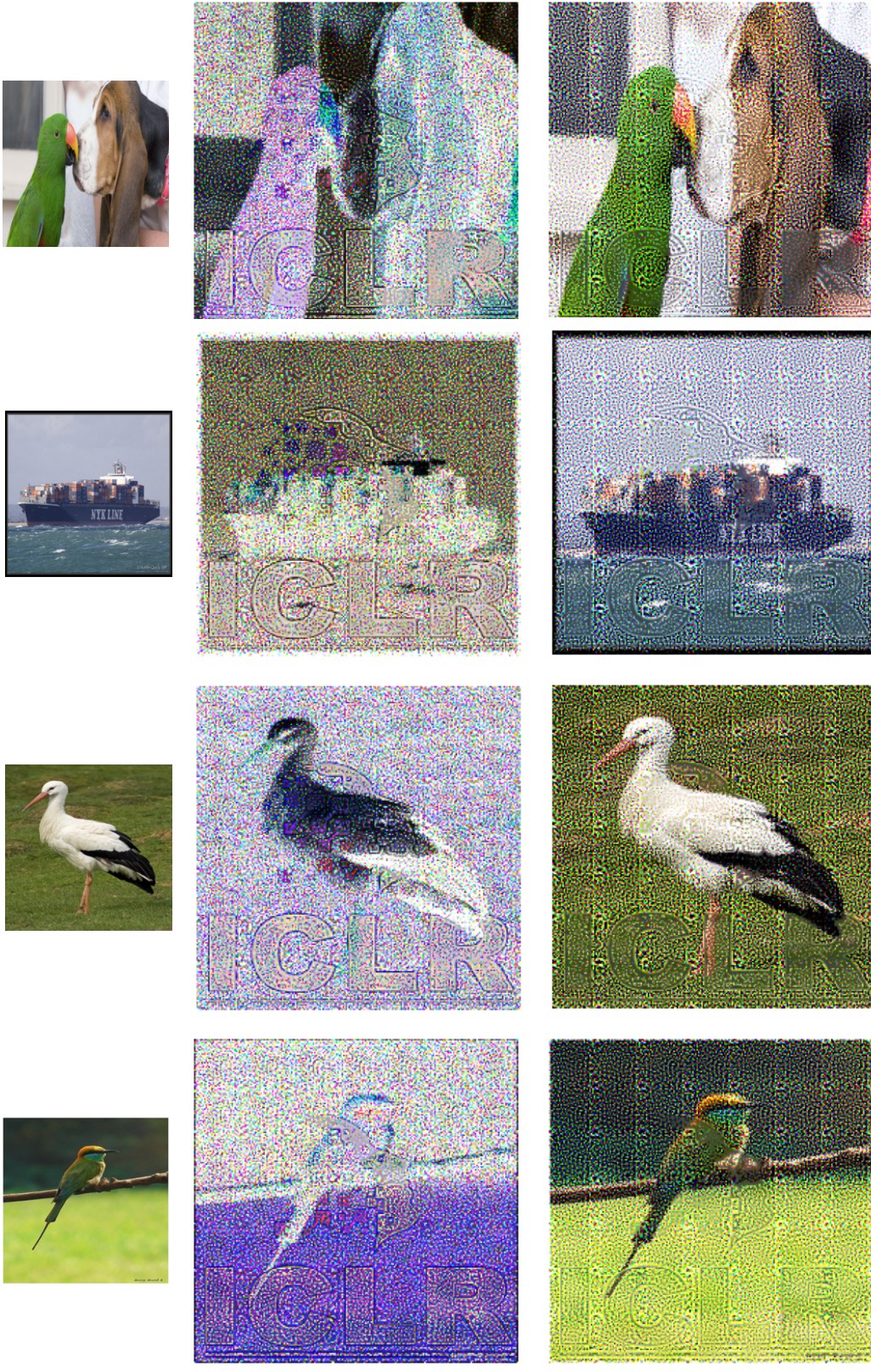

Figure 6: **Watermark superposition using the nullspace basis vectors.**

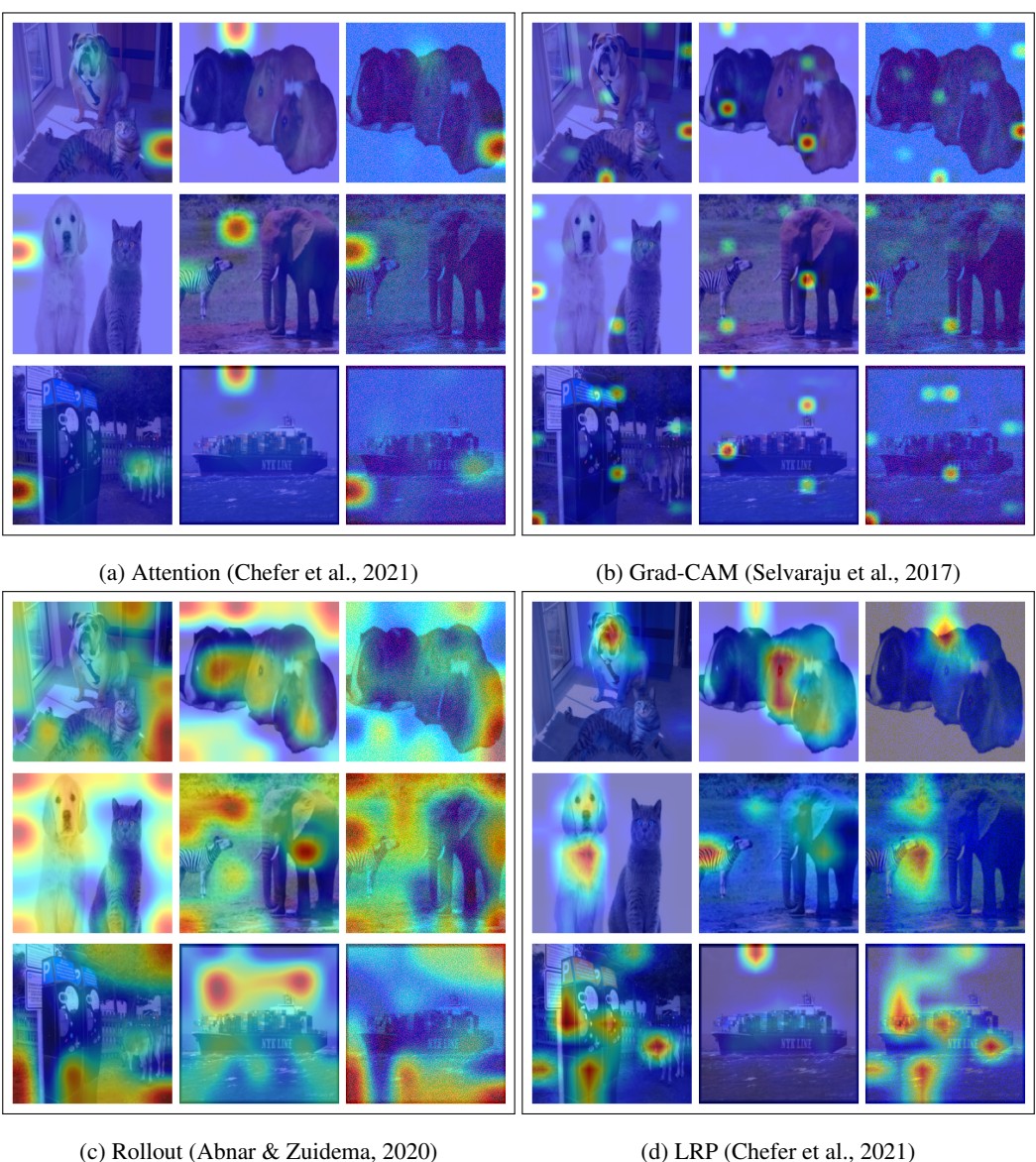

(a) Attention (Chefer et al., 2021)

(b) Grad-CAM (Selvaraju et al., 2017)

(c) Rollout (Abnar & Zuidema, 2020)

(d) LRP (Chefer et al., 2021)

Figure 7: **Interpretability maps generated via different methods for (source, target, transformed) images**

