# OpenReview forum: "On Nullspace of Vision Transformers and What Does it Tell Us?"
_ICLR.cc/2023/Conference — Submitted to ICLR 2023_

### Official Review · Reviewer_kZts · 2022-10-24

**Confidence:** 4
**Correctness:** 3
**Technical Novelty And Significance:** 4
**Empirical Novelty And Significance:** 4
**Recommendation:** 5

**Clarity, Quality, Novelty And Reproducibility:**


Clarity
- In the introduction section, the explanations of background and  ViT are long, and the number of references (6 pages) seems too large. The authors can reduce the number of papers for the introduction sections.

- Figure 1 left bottom equation: What means $\eta$, $\beta_1$, $\beta_2$ is unclear.

- Figure 1 caption. "right)" seems to be a typo.

Quality

The application is evaluated only with the visualization of a few images.
These applications might be evaluated quantitatively, e.g. recognition accuracy for the unconstraint targeted noise.

Novelty

The introduction of nullspace for analyzing ViT is novel. The two applications are also novel and interesting.


**Strength And Weaknesses:**

Strength

- The introduction of nullspace for analyzing ViT has originality. This paper investigates if the nullspace exists for the linear input layer (patch embedding layer) and non-linear layer (self-attention layer). The patch embedding layer's linear structure produces a larger nullspace than CNNs, as shown in Sec.5.2.

- The watermark application and the targeted nullspace noise are interesting. They are largely different from conventional adversarial attacks.

Weakness
- The results of watermark supervision and targeted nullspace noise are not visually attractive.

- Writing quality is not high.


**Summary Of The Paper:**

This paper explores the nullspace ( adding an element on this space does not affect the output) for Vision Transformer. The authors first demonstrate that a non-trivial nullspace exists for the patch embedding matrices. This idea is extended to the non-linear layers of the vision transformer, which is learned via simple optimizations.
The applications to image watermarks and unconstrained targeted noise are presented.

**Summary Of The Review:**

This paper presents a nullspace analysis of ViT, which have originality. The two applications of nullspace are also interesting. Though it is understandable, the writing and evaluations are not high quality.

---

> ### Author Response · Authors · 2022-11-07
> **Thank you for the feedback!**
>
> We'd like to thank the reviewer for their time and effort.
>
> > The results of watermark supervision and targeted nullspace noise are not visually attractive.
>
> These examples served to display the potential of nullspace noise. Though not perfect these still highlight significantly different ideas from existing work on the noise synthesis for example adversarial images. Targeting individual applications and improving them further will be the focus of our future work.
>
> > Figure 1 left bottom equation: What means $\nu, \beta_1, \beta_2$  is unclear.
>
> $\beta$s are the parameters of a linear function. $\nu$ is the noise sampled from $\mathbb{R}$. We portrayed the concept of nullspace in terms of the linear function using these variables.
>
> > Figure 1 caption. "right)" seems to be a typo.
>
> $\textit{left-bottom, left-top, right}$ refer to the $3$ different scenarios in which the nullspace is highlighted in figure 1. This is not a typo.

---

### Official Review · Reviewer_upQH · 2022-10-25

**Confidence:** 5
**Correctness:** 3
**Technical Novelty And Significance:** 2
**Empirical Novelty And Significance:** 2
**Recommendation:** 3

**Clarity, Quality, Novelty And Reproducibility:**

The paper is well-written. The quality and novelty are not high. There is no submitted code to reproduce the results.


**Details Of Ethics Concerns:**

I have no ethics concerns for this paper.


**Strength And Weaknesses:**

**Strong points:**

1. The paper addresses an interesting problem as finding the nullspace of ViT will tell us certain types of input perturbations to which the network is inherently robust.

2. The paper is well-written with illustrative figures.

**Weak points:**

1. In the nonlinear case where the nullspace is ill-defined, the authors “attempt to preserve the axiom of most interest to us, closeness under vector addition” and look for an alternative set that adds any element in that set to the input, and the output of the transformation remains unchanged. This approach lacks theoretical justification.

2. The relaxation to study an individual element of the nullspace in the nonlinear case (e.g. when deriving the nullspace for the self-attention stage) removes all interesting insights from studying the whole nullspace of ViT and makes the contribution of the paper mediocre.

3. For noise robustness application, I have a concern about the significance of the idea. Finding the nullspace noise only allows us to know a certain set of noise to which the ViT is robust. How does it help improve the generalization of ViT models? How to constrain other input permutations/corruptions to be in the approximated nullspace of the network?

4. For the applications relating to watermarking, the nullspace-noise-added images significantly decrease the quality of the original images, making them not useful in reality. For the applications on output-preserved image watermarking, the quality of the corrupted images is bad and much worse than the adversarial examples generated by adding adversarial noise into the original images.

**Additional Concerns and Questions for the Authors:**

1. $\tilde{v}\_{\phi}$ is learned by minimizing Eqn. 6. I am confused whether the authors learned one $\tilde{v}\_{\phi}$ or a set of $\tilde{v}\_{\phi}$ for each self-attention stage. In the first case, $\tilde{v}\_{\phi}$ will be input-independent, then it will no longer be a noise added to the input.

2. The Eqn. 2 in the paper is not correct. \Delta should be a subspace, not vectors.

3. There is no discussion of related work on the nullspace of transformers, such as the work in [1].

**References:**

[1] Brunner, Gino, Yang Liu, Damian Pascual, Oliver Richter, Massimiliano Ciaramita, and Roger Wattenhofer. "On identifiability in transformers." arXiv preprint arXiv:[1] Brunner, Gino, Yang Liu, Damian Pascual, Oliver Richter, Massimiliano Ciaramita, and Roger Wattenhofer. "On identifiability in transformers." arXiv preprint arXiv:1908.04211 (2019). (2019).



**Summary Of The Paper:**

The paper explores the concept of nullspace for vision transformers (ViTs). The authors find that there usually exists non-trivial nullspace for vision transformers by finding the exact nullspace of their patch-embedding stages. Extending to the case of non-linear transformation where the nullspace is ill-defined, the authors look for an alternative set that adds any element in that set to the input, and the output of the transformation remains unchanged. The authors also study the robustness of ViTs with the nullspace noise perspective. In addition, they show that ViTs are more robust than CNN models under the noise generated via the nullspace. The authors also illustrate that nullspace noise can be used to impose perceptible watermarks on images.


**Summary Of The Review:**

Overall, I vote for rejecting. The idea of approximating the nullspace noise of the Vision Transformer (ViT) model is interesting. However, my main concerns about the paper are: 1) the low significance of the proposed method to study the nullspace of ViTs in the paper, 2) the lack of theoretical justification of the proposed method in the nonlinear setting, and 3) the unconvincing applications of the proposed method.

---

> ### Author Response · Authors · 2022-11-07
> **Author response (1/2)**
>
> Thanks for spending time and effort on the review and providing us with invaluable feedback! We are happy that you found the problem of exploring nullspace of ViTs interesting.
>
> > In the nonlinear case where the nullspace is ill-defined, the authors “attempt to preserve the axiom of most interest to us, closeness under vector addition” and look for an alternative set that adds any element in that set to the input, and the output of the transformation remains unchanged. This approach lacks theoretical justification.
>
> Due to non-linearity of the mapping we drop the pursuit of a well-defined subspace of the inputs to seek all sets of nullspace vectors. The reason we chose `closeness under vector addition` is because it allows us to learn a noise vector which can be applied to any input without affecting the output of the network. This falls nicely into the definition of additive noise which we introduced in the previous section of Exact nullspace. The axiom of `zero` element provides us a trivial solution. The axiom of `closeness under multiplication` does not give us a noise vector as the one we obtain from the exact nullspace.
>
> >  The relaxation to study an individual element of the nullspace in the nonlinear case (e.g. when deriving the nullspace for the self-attention stage) removes all interesting insights from studying the whole nullspace of ViT and makes the contribution of the paper mediocre.
>
> We believe a study comparing robustness of different networks to any kind of noise is a very interesting problem. It still remains interesting even with the approximated noise as well. We now know that it is easier to find nullspace noises for the ViTs than CNNs (at enc or input level). Moreover, we can always club the exact nullspace noise with the approximated noise to still be able to synthesise infinite noises. In this paper we aimed to laying the ground work for nullspace noises and its application. In the future, we would like to investigate the feasibility of learning an underlying noise generating process.
>
> > For noise robustness application, I have a concern about the significance of the idea. Finding the nullspace noise only allows us to know a certain set of noise to which the ViT is robust. How does it help improve the generalisation of ViT models? How to constrain other input permutations/corruptions to be in the approximated nullspace of the network?
>
> In this work, we do not aim to unify the corruptions in terms of the nullspace noise, and instead focus on serving an exploratory study to highlight the not so explored nullspace of ViTs. At first glance we don't think nullspace can explain general robustness to other corruptions, very much like adversarial robustness provides no indication of robustness to other types of corruptions. However, the connection between nullspace noise and other noise types is a very interesting question as raised in the review. We believe that learning to constraint other corruptions to the nullspace in order to improve robustness is a very challenging problem, and it should be tackled as a standalone research direction.
>
> > For the applications relating to watermarking, the nullspace-noise-added images significantly decrease the quality of the original images, making them not useful in reality. For the applications on output-preserved image watermarking, the quality of the corrupted images is bad and much worse than the adversarial examples generated by adding adversarial noise into the original images.
>
> We'd like to recall that the aim of the perceptible watermarking process is not to preserve the original quality of the image. The main reason behind watermarking an image is to discourage unapproved distribution and utilisation of the source content. A perceptible watermark is thus placed which always deteriorates the original image quality (for example [Shutterstock](https://www.shutterstock.com/image-photo/rakotz-bridge-rakotzbrucke-devils-kromlau-saxony-758202754) images).
> Adversarial images have a completely different use case than watermarked images. Typically, it is preferred that the adversarial images don't look significantly different from the original image while being misclassified by a network. Whereas, watermarked images are meant to signal the authorship and protective nature of the content. These two sets of synthesised images serve two very different purposes.
>
>  Nullspace noise preserves the model's behaviour on the modified input even with severe degradation. Hence watermarking here serves two purposes a) generates identical model behaviour for watermarked image as an original image b) The degradation of the input here in turn discourages storage and distribution of a user's content without permission.

---

> > ### Comment · Reviewer_upQH · 2022-11-19
> > **Reply to the Author’s Rebuttal**
> >
> > Thanks the author for the response. Please find below my concerns and questions after reading the author’s rebuttal.
> >
> > 1. We agree with the reviewer that watermarking is to discourage unapproved distribution and utilization of the source content. However, the quality of the watermarked images in Figure 4 and 6 are bad, which make the proposed method not useful at all for this application.
> >
> > 2. Regarding noise robustness applications, many methods have been proposed to improve the robustness of transformers across multiple types of realistic perturbations [1,2,3], not only under the null space noise. Thus, I still question the usefulness of the null space approach proposed in the paper for improving the robustness of transformers.
> >
> > Since Discussion Stage 1 is almost ended. You can include the answers and results for my questions without the need to put them in the manuscript. I am looking forward to more discussion in Discussion Stage 2.
> >
> > **References**
> >
> > [1] Bhojanapalli, Srinadh, Ayan Chakrabarti, Daniel Glasner, Daliang Li, Thomas Unterthiner, and Andreas Veit. "Understanding robustness of transformers for image classification." In Proceedings of the IEEE/CVF International Conference on Computer Vision, pp. 10231-10241. 2021.
> >
> > [2] Mao, Xiaofeng, Gege Qi, Yuefeng Chen, Xiaodan Li, Ranjie Duan, Shaokai Ye, Yuan He, and Hui Xue. "Towards robust vision transformer." In Proceedings of the IEEE/CVF Conference on Computer Vision and Pattern Recognition, pp. 12042-12051. 2022.
> >
> > [3] Chen, Jie-Neng, Shuyang Sun, Ju He, Philip HS Torr, Alan Yuille, and Song Bai. "Transmix: Attend to mix for vision transformers." In Proceedings of the IEEE/CVF Conference on Computer Vision and Pattern Recognition, pp. 12135-12144. 2022.

---

> ### Author Response · Authors · 2022-11-07
> **Author response (2/2)**
>
> > $\tilde{v}_\phi$ is learned by minimizing Eqn. 6. I am confused whether the authors learned one
>  or a set of for each self-attention stage. In the first case, will be input-independent, then it will no longer be a noise added to the input.
>
> We learn one noise for the entire backbone and not for each attention step. Yes, it is independent of the input and is a property of the neural network instead. I still remains an additive noise which is universal for the model and is ignored by it when added to $any$ input.
>
> > There is no discussion of related work on the nullspace of transformers, such as the work in [1].
>
> Thank you for bringing the above referred work to our notice. We have added it to the discussion in our work. To highlight the differences, [1] computes nullspace of the self-attention weights to argue against the identifiability of attention. Though termed weights these are essentially responses which vary per input. Different from this setting, we explore the nullspaces of the weights (parameters) of the ViTs which allows the study to be input independent and provides significantly different applications and insights.

---

### Official Review · Reviewer_ZgDX · 2022-10-25

**Confidence:** 4
**Correctness:** 4
**Technical Novelty And Significance:** 4
**Empirical Novelty And Significance:** 3
**Recommendation:** 5

**Clarity, Quality, Novelty And Reproducibility:**

The paper was very well written and was very easy to follow. The authors did not mention that the code will be released upon publication. The exploration of nullspace noise for analyzing the robustness of models is a novel application of the concept.

**Strength And Weaknesses:**

Pros
	- The exploration of nullspace noise for analyzing the robustness of models is a novel application of the concept. I would love to see more exploration into the implications of nullspace noise on robustness.
	- The paper was very well written, and easy to follow.

Cons
	- The paper mainly discusses 3 areas: robustness, watermarking and adversarial generation.
		§ Robustness:
			§ I think this area requires more analysis in the paper. The introduction and conclusion sections imply that the existence of nullspace vectors, and the high amount of noise generated using this nullspace basis, has significant implications on the robustness of the vision transformer.
			§ Section 5 presents the question: "How robust are different architectures to nullspace noises." However, this seems a bit confusing. My understanding is, all architectures are, by definition, robust to their nullspace noises. Further, each network has its own nullspace noise distribution, depending on its architecture. The main question, I think, is more about the properties of the nullspace noises of a given model. A model is more robust if:
				□ Its nullspace has a high sampling limit (with a high threshold on accuracy or %match)
				□ The nullspace samples, as described by equation (4), have high accuracy too.
			§ Figure 3 (a) and (b) show that ViT have higher sampling limits than state-of-the-art CNNs. I think these results give evidence for the first property of robustness w.r.t nullspace noise as described above. However, I think it will be really helpful to analyse these results on more datasets, instead of just a single one.
			§ Figure 3(c) shows low transferability of noise between models, however, I don't think this particular graph makes any implications on which model is better. Since the noise vectors are obtained using ViT-S model, they are strongly biased against the CNNs. It's very likely that the ViT models will perform equally poorly on the noise vectors obtained using the CNNs. I think it would be helpful if this point is highlighted.
			§ Figure 3 doesn’t include any quantitative results for the 2nd property described above, regarding the accuracy of nullspace samples from equation (4). This can probably be done by presenting the deviation of accuracy at different nullspace noise vectors, and increasing the number of sampling limits considered.
			§ It would be helpful if the point about "lack of effective nullspace noise" in CNNs is elaborated a bit more. Does the difficulty in finding effective noise due to non-linearity necessarily mean lack of effective nullspace noise?
		§ Watermarking:
			§ In this application, a slightly different watermark will be applied on each image. It would be helpful if the implications of this are discussed. Would this significantly comprise the tasks for which the watermarks are being used for? I am curious to see if this has any practical implications.
		§ Adversarial Sample Generation:
			§ I think more results are required for this application. It would be really helpful if the following results are included:
				□ An accuracy score on a few datasets on the misclassifications, along with the qualitative examples, shown in Figure 5 (a).
				□ It would be helpful if the explainability attribution map mistakes are quantified in some manner, probably IoU/DICE etc, to get some quantitative results to support the qualitative results shown in Figure 5 (b).

Minor Edits
	- In section 2, paragraph starting with "A trivial nullspace, …", shouldn't "axiom 1" be used instead of "linear mapping as described in equation 1", and "equation 2" instead of "linear equations as described by equation 1"?
	- In section 3.1 Classification Stage, shouldn't it end with "project it through a linear classification layer, followed by a softmax layer" instead?
	- In equation (5), it would be more clear to add f_phi instead of just f.
	- Section 6 method details: typo at delta x_j -> "changed" to "change"
In section 7 line 3, helpful to write equation (7) instead of just 7.


**Summary Of The Paper:**

The paper presents an exploration of the concept of nullspaces for vision transformers. It presents an approach towards determining the nullspace vectors of the vision transformer: a deterministic non-trivial nullspace for linear projection components and an approximate nullspace for non-linear attention layers. Further, it explores its implications in 3 areas: robustness, watermarking and adversarial sample generation to fool models and interpretability methods. Finally, the paper shows that ViTs are more robust to higher amounts of nullspace-generated noise than state-of-the-art CNNs, and demonstrates the application of nullspace noises in watermark imposition and adversarial sample generation.

**Summary Of The Review:**

Borderline reject. The paper definitely proposes a novel idea, however, I think a bit more analysis and results are required. If the author(s) are able to address the above comments, the decision will be reconsidered.

---

> ### Author Response · Authors · 2022-11-07
> **Author response**
>
> Thank you for your valuable feedback!
>
> > Section 5 presents the question: "How robust are different architectures to nullspace noises." However, this seems a bit confusing.
>
> As pointed out in the review, for exact nullspace noise, a network will always remain invariant to the added noise. However, for approximate nullspace noise, some networks perform better than others for range of sampling limits. We have modified the section heading to state that it is $\textit{approximate nullspace noise}$ which we are comparing for each network.
>
> > Figure 3(c) shows low transferability of noise between models, however, I don't think this particular graph makes any implications on which model is better. Since the noise vectors are obtained using ViT-S model, they are strongly biased against the CNNs. It's very likely that the ViT models will perform equally poorly on the noise vectors obtained using the CNNs. I think it would be helpful if this point is highlighted.
>
> Duly noted. We intended to emphasise that nullspace noise is not transferable. We have now explicitly mentioned that transferability is expected to be low from CNN to ViT and other way around as well. In summary, nullspace noise is local to a particular model and does not transfer as it is the case for adversarial samples.
>
> > Figure 3 doesn’t include any quantitative results for the 2nd property described above (The nullspace samples, as described by equation (4), have high accuracy too) , regarding the accuracy of nullspace samples from equation (4). This can probably be done by presenting the deviation of accuracy at different nullspace noise vectors, and increasing the number of sampling limits considered.
>
> The nullspace samples don;t have high accuracy. The accuracy post noise will match the accuracy of the model without noise. For example, the accuracy of a typical ViT on Imagenet is 76\% the accuracy of the network after introduction of noise will be 76\% as well.
> Moreover, this observation will be true regardless of the dataset the network is evaluated on. However, for approx. nullspace noise which is sampled at different limits the drop in accuracy will follow the trend of drop in match\% performance. For example, 100\% match percentage reflects a 76\% validation score which decreases as the match score decreases towards larger limit.
>
> > It would be helpful if the point about "lack of effective nullspace noise" in CNNs is elaborated a bit more. Does the difficulty in finding effective noise due to non-linearity necessarily mean lack of effective nullspace noise?
>
> We believe that as per the performance, purely CNN architectures demonstrate rather low tolerance to nullspace noise. Unlike ViTs where an exact nullspace exists at the very first layer, it is not feasible to analytically determine for purely CNN architectures. We have added 2 more architectures for the evaluation Swin and Convnext. Both these networks also exhibit high tolerance to nullspace noise ($\tilde{v}_\omega$)for a wide range of `lim`. Swin and convnext both utilise patch embeddings similar to ViTs but they fall into transformer based and CNN based architectures. A simple patch embedding layer according to us makes the difference in approx. nullspace noise robustness.
>
> > Adversarial Sample Generation: § I think more results are required for this application. It would be really helpful if the following results are included: □ An accuracy score on a few datasets on the misclassifications, along with the qualitative examples, shown in Figure 5 (a).
>
> The beauty of nullspace noise is that it is the property of the network independent of the dataset it is trained on or applied to. The classification rate (comparing source and transformed images) or the misclassification rate (comparing target and transformed images) will be always 100\% and 0\% if we utilise the exact nullspace noise from a model.
>
> > It would be helpful if the explainability attribution map mistakes are quantified in some manner, probably IoU/DICE etc, to get some quantitative results to support the qualitative results shown in Figure 5
>
> For Trans-LRP, the IoU (source, transformed) and (target, transformed) is 90.22\% and 2.5\% respectively. We used half the max value for transformed heatmap as the threshold to binarize the images prior to IoU computation. The quantitative results agree with the qualitative observations that the heatmaps of transformed images depict high similarity to source images than target images. We have added this observation to the supplementary and mentioned the same in the main paper.
>
> > The authors did not mention that the code will be released upon publication.
>
> Though we shared all the details for replicating the results, we will also release the code.
>
> `Lastly, we have fixed the minor corrections as pointed out in the review.`

---

> ### Author Response · Authors · 2022-11-08
> **Results on more datasets**
>
> We  report results of a quick experiment (figure 3.b) on CIFAR-10 and STL-10 dataset for Resnet-20, Mv2 and Vit architectures.
>
> `CIFAR-10`
>
> | lim  | 0.01  | 0.05  | 0.1  | 0.25  | 0.5  | 1.0  | 1.5  | 2.0  |
> |---|---|---|---|---|---|---|---|---|
> | Vit  |  28.69 | 30.81  | 30.36  | 28.30  | 29.26  | 25.73  | 25.39  | 20.02  |
> | R-20  |  21.12 | 22.84  | 19.49  | 18.95  | 16.72  | 10.46  | 10.53  | 10.66  |
> |  Mv2 |  26.84 | 24.30  | 25.32  | 24.08  | 24.25  | 16.26  | 17.86  | 11.08 |
>
> `STL-10`
>
> | lim  | 0.01  | 0.05  | 0.1  | 0.25  | 0.5  | 1.0  | 1.5  | 2.0  |
> |---|---|---|---|---|---|---|---|---|
> | Vit  |  99.24 | 98.64  | 98.36  | 97.41  | 92.30  | 79.44  | 72.86  | 64.34  |
> | R-20  | 86.86  | 85.05  | 81.35  | 69.90  | 49.02  | 39.75  | 36.14  | 32.34  |
> |  Mv2 | 89.13  | 87.33  | 83.17  | 74.06  | 59.61  | 35.56  | 32.45  | 29.14 |
>
> The results indicate a similar trend that we observed for Imagenet trained models. ViTs are able to maintain high resistance through different limits.

---

### Official Review · Reviewer_ZuY5 · 2022-10-31

**Confidence:** 4
**Correctness:** 2
**Technical Novelty And Significance:** 2
**Empirical Novelty And Significance:** 2
**Recommendation:** 3

**Clarity, Quality, Novelty And Reproducibility:**

**Clarity**: The paper is mostly clear. There are some easily fixable typos.

**Quality**: Should be improved. I feel that the technical portion of the paper is not rigorous enough. The parallel between the concept of linear null spaces and that of a single additive perturbation to the input is stretching the concept too much. The technical framework appears artificial. Also, experimental evaluation needs to be improved.

**Novelty**: Though I’m not aware of any work that analyzes the null space of patch embeddings in ViT, it is straightforward and well-known in classical literature. The extension to nonlinear parts of the network would be of interest but is ad hoc and does not add much value to the field.

**Reproducibility**:  Should be reproducible.


**Strength And Weaknesses:**

## Strengths
- Investigation of null spaces of vision transformers is a good direction of research and of relevance to the ICLR and wider AI/ ML community.
- There are some interesting ideas regarding finding perturbations in the input space to which the output of ViT is invariant and to use it for perceptible watermarking.

## Weaknesses

__(W.1) Importance__: The motivation and importance is not quite clear. The authors make an argument about inherent robustness (invariance) to certain perturbation in the input. It is quite obvious that by design, classification networks are sought to be invariant to tp intra-class differences and geometric and photometric transformations that are deemed not to result in changes in the class identity, leading to all kinds of data augmentations. That set of subspaces of inputs may lie in the nullspace is the key idea here but it is not clear what nontrivial value the authors provide in the paper. Please elaborate.

 __(W.2) Technical Approach__: I have several questions regarding the technical approach proposed in Section 4  (if I’m understanding it correctly):
- __(W.2.1)__: Equation (5) describes a set which is invariant to the input $\mathbf{u}$. However, the loss functional defined in equation (6) doesn’t reflect his invariance to the input $\mathbf{u}$. Is there an empirical expectation over the entire training dataset?
 - __(W.2.2)__: The loss in Equation (6) admits a trivial solution ($\tilde{\mathbf{v}}_\phi = 0$. There is no regularization which keeps the solution away from the trivial one. Even if empirically the minimization (perhaps through SGD) yields a solution that’s nontrivial, it can still be arbitrarily close, lying in a small non-isotropic neighborhood around $\mathbf{u}$ with the same output value. The non-isotropic nature (think elongated tubes, for example) will allow a random permutation of the perturbation to change the output. The existence of this small neighborhood is just a statement about local regularity (of data spaces) and smoothness (of functions) on it. How is this useful in the sense that you seek to exploit in the paper?
- __(W.2.3)__: The solution of Eqn. (6) resulting in a  'nullspace' noise vector which is not arbitrarily close to __any input u__ while maintaining the same prediction as __u__ for every __u__ (i.e., it is independent of __u__) is not demonstrated or validated. At least on the training data, the % match predictions should be 100% for the definition to make sense. Is this the case?
- __(W.2.4)__: It seems like a single noise vector is learnt over the entire training dataset per training episode (for each random initialization of the noise vector $\tilde{v}_\phi$). This is rather inefficient for generating ‘infinite such perturbations’ as claimed in the motivation. Secondly, it’s unclear what the diversity of the set of noise vectors is.
- __(W.2.5)__:  Magnitude(v) should be shared and the significance of MSE(logit) should be explained. It is not clear what to make of the numbers on the y-axis (0-7000.0). How large are those numbers with respect to the magnitudes of the logits?
- __(W.2.6)__:  For classification networks, simple geometric transformations like translations, rotations, small scalings etc. should lead to the same classification output. How are the random perturbations found through equation (6) useful beyond the above?
- __(W.2.7)__:  The whole discussion in 5.2 about strided convolutions and intractability of determining a null space seems mathematically flawed (if we ignore the pointwise nonlinearity of the ReLU etc. which the authors don’t seem to be considering) since tensor multiplication is a linear operation. Kindly explain. [1’] below may be useful to take a look at.

[1’] Zou, Dongmian, Radu Balan, and Maneesh Singh. "On lipschitz bounds of general convolutional neural networks." IEEE Transactions on Information Theory 66.3 (2019): 1738-1759.

 __(W.3) Experimental Evaluation__:
- __(W.3.1) Robustness__: In Figure 3(a), why should the scale of the features with respect to which the sampling limit is meaningful, be the same?
- __(W.3.2) Robustness__: Comparison of robustness to noise across different networks makes sense when the performance degradation under the same perturbation of the input is analyzed. In the experiment in Figure (3), the learnt perturbation also varies and is from an optimization algorithm (which may fail to do what’s expected of it). In this ‘approximate’ setting, everything is so amorphous and the different factors are so entangled that it’s hard to have any takeaways from this experiment. Kindly explain.
- __(W.3.3) Watermarking__: Equation (7) suggests that the transformed image should look like the target (watermark) and yet have the same classification semantics as the source. However, the transformed images in Figure (4) look more like the source images while images in Figure (5) look more like the target images. It is not clear what the goal of visible watermarking is and what kind of visual perturbations are considered to be of acceptable quality and for what purposes. Kindly clarify.
- __(W.3.4) Watermarking__: Qualitative results provided in A.2 should be in the main paper.
- __(W.3.5) Watermarking__: Only twenty images are used for evaluation. This is unsatisfactory. Since only the null space of the patch embeddings is considered ($v_\theta$), why should this be prohibitively expensive?
- __(W.3.6) Watermarking__: Finally, since by design, $v_\theta$ should not alter the representations by the network, should the classification accuracy be higher than 85%, and 100% if training images are used?
- __(W.3.7) Unconstrained Targeted Noise__: No quantitative results are provided.


**Summary Of The Paper:**

The paper proposes an approach to identify the null space of vision transformers (ViT). They divide this into two parts – (a) null space of patch embeddings, and, (b) null space for the self-attention module(s). The paper uses the above to study the robustness of ViT to null space noise, compare the impact on different architectures, and study its transfer properties. Finally, the null-space is used to apply perceptible watermarks to images without altering the test-time performance (of image classification).


**Summary Of The Review:**

I’m not convinced of the merits of the paper. There are some interesting ideas. However, the technical framework appears artificial, somewhat ad hoc, and not rigorous enough. More extensive experimental validation is required. As a result, I don’t assess the current submission to be of publishable quality.

---

> ### Author Response · Authors · 2022-11-07
> **Author response (1/2)**
>
> Thank you for your valuable feedback!
>
> `(W.1), (W.2.6)` The noise as introduced in this work is altogether a different family from well known augmentations currently in practice. The nullspace noise comes with a stronger guarantee that a network is robust towards noise sampled from its nullspace. This is not always the case in terms of traditional augmentations where a network can arbitrarily exhibit unexpected behaviour (wrong predictions due to change in brightness, contrast etc). We are not introducing a new augmentation to be used in training, we simply explore robustness of ViTs to a new family of perturbations.
> The non-trivial nullspace values in terms of magnitude are visualised in the perceptible watermarking and targeted nullspace noise sections (Figure 4.c)
>
> `(W.2.1)` Yes, there is an empirical expectation over the entire training dataset. Now fixed.
>
> `(W.2.2)` Yes, there is no explicit regularisation added to the optimisation. For an explicit regularisation such as on norm of the noise values we require double backward propagation thorough the network. We found that doing this yields similar quantitative results but while taking considerably more time. We believe the reason why it works without regularisation on learnt noise is due to high non-linearity of the learning process. By the argument presented in the paper for closeness to $\mathbf{u}$, the learnt noise vector $\mathbf{v}$ then needs to lie close to $\textit{every}$ input sample. Moreover, we demonstrated that by permuting $\mathbf{v}$ the match\% decreases considerably. For a good learnt noise $v$ we wish $f(\mathbf{u}+\mathbf{v}) = f(\mathbf{u})$ and $\mathbf{v}\neq\mathbf{0}$.
>
> `(W.2.3)` We are not aiming to make $\mathbf{v}$ be different from every input $\mathbf{u}$. Only the final predictions ($f(\mathbf{v}+\mathbf{u}) = f(\mathbf{u})$) should remain invariant under the learnt noise. We demonstrated this using the match percentage and mse metric. Due to the approximate nature of $\mathbf{v}$ the match percentage is never 100\%, even on the training data. For exact nullspace noise, the match\% will always be 100 no matter the input data.
>
> `(W.2.4)` The approximate nullspace noise is indeed learnt once per episode. However, to induce variation, one can always add exact nullspace noise from the patch embedding layer. Improving efficiency of approximate nullspace noise sampling is something we would like to pursue in our future work.
>
> For diversity we share below the average abs value of the approx nullspace noise ($\tilde{v}_\omega$)
>
> |  lim | 0.01  | 0.05  | 0.1  | 0.25  | 0.5  | 1.0  | 1.5  | 2.0  |
> |---|---|---|---|---|---|---|---|---|
> |  r50 | 2.02  | 1.95  | 1.90  | 1.87  | 2.04  | 2.93  | 4.48  | 5.91  |
> |  vit-b | 2.10  | 1.98  | 2.00  | 2.10  | 2.33  | 2.89  | 3.73  | 4.69  |
>
> Note that even though the avg noise is lower for Resnet-50 for few sampling lim, the match \% for ViT is always larger. We will add the plot corresponding to mean with std to highlight the range of values the approximated noise can take.
>
> `(W.2.5)` The logits are simply the classification output prior to the softmax normalisation. The reported mse is summed over the 1000 categories of ImageNet and then averaged over all the validation samples. It is provided to compare relative errors of different architectures minimising the same objective.
>
> `(W.2.6)` We aimed to highlight an argument for ViT vs CNNs using approx nullspace noise which is of interest to the computer vision community. Moreover, we believe that learning better approx nullspace noise will help in improving watermarking and targeted nullspace noise applications which right now is limited to exact nullspace noise.
>
> `(W.2.7)` The discussion is around learning the nullspace noise directly at the input level utilising the convolution kernels from the very first layer of a CNN. Convolution is indeed a linear operation, however, to learn an input noise the identified nullspace noise needs to be shared by all the kernels at the first layer. Moreover, as the convolutions are overlapping this imposes an additional constraint on the input noise. By being intractable we meant that it is not possible to obtain the subspace as is the case with ViTs.

---

> ### Author Response · Authors · 2022-11-07
> **Author response (2/2)**
>
>
> `(W.3.1),(W.3.2)` With 3.a and 3.b we show two different scenarios of the induced noise. For (a), we learn the noise at the non-linear level. In (b), the noise it learnt at a *common input level*.  The level does not have to be the same, for this reason we provided learning noise at the input level as well.
>
> `(W3.3)` Yes, the objective is indeed to transform the entire input. However, we observed that the constraint over the values (0-255) and utilising only linear combinations of the nullspace basis vectors, the conversion process is incomplete and results in preserving partial information from both source and target. Alternatively we can manually set the target as a linear combination of input and the watermark, which yields similar qualitative results. However, in this case the watermarked response will not be equal to the original image but rather the blended target image.
>
> `(W.3.4), (W.3.6)` The qualitative results are in the main paper (fig 4), we have also rectified the error in quantitative results. The match percentage is 100\% with 0.01\% difference in the final predicted confidence. We'd like to point out that the match is expected to be 100\% for $v_\theta$ regardless of training or validation images.
>
> `(W.3.5)` The watermarking optimisation is required to be executed for each image separately. The small number of images is intended to support the argument of 100\% performance retention under nullspace noise. Even after applying the watermarked noise from one image to all images we obtain 100\% performance retention regardless of the training/validation data.
>
> `(W.3.7)` For Trans-LRP, the IoU (source, transformed) and (target, transformed) is 90.22% and 2.5% respectively. We used half the max value for transformed heatmap as the threshold to binarize the images prior to IoU computation. The quantitative results agree with the qualitative observations that the heatmaps of transformed images depict high similarity to source images than target images. We have added this observation to the supplementary and mentioned the same in the main paper.

---

> > ### Comment · Reviewer_ZuY5 · 2022-11-28
> > **Acknowledgment**
> >
> > Thanks a lot for the response. I have gone through the response and factored that in my review.

---

### Decision · Program_Chairs · 2023-01-20

**Decision:**

Reject

**Justification For Why Not Higher Score:**

The paper results are preliminary and reviewers note the following weaknesses

1) The proposed approach to find null space vectors for nonlinear models lacks proper theoretical grounding and justification.
2) Not a clear application to improving robustness of the networks.
3) Reduced quality for the proposed watermark application.


**Justification For Why Not Lower Score:**

N/A

**Metareview: Summary, Strengths And Weaknesses:**

All the reviewers find the problem studied, on the null space of neural networks, interesting. However they find the current draft to be preliminary and not sufficiently developed. In particular they note the following limitations.

1) The proposed approach to find null space vectors for nonlinear models lacks proper theoretical grounding and justification.
2) Not a clear application to improving robustness of the networks.
3) Reduced quality for the proposed watermark application.

Authors provide more intuition and some quantitative metrics during the discussion stage, but the fundamental weakness remain and all the reviewers remain unconvinced of the current draft. Hence I suggest rejection.